# Estimation of Transition Frequency during Continuous Translation Surface Perturbation

**Nur Fatin Fatina Mohd Ramli [1,\*], Mohd Azuwan Mat Dzahir [1,2] and Shin-Ichiroh Yamamoto [1]**

[1]  Department of Bioscience Engineering, Shibaura Institute of Technology, 307 Fukasaku, Minuma-ku, Saitama-City, Saitama 337-8570, Japan; azuwan@utm.my (M.A.M.D.); yamashin@se.shibaura-it.ac.jp (S.-I.Y.)

[2]  School of Mechanical Engineering, Faculty of Engineering, Universiti of Teknologi Malaysia, Skudai 81310, Johor Bahru, Malaysia

\*  Correspondence: nb16106@shibaura-it.ac.jp; Tel.: +81-80-4084-2365

**Abstract:** Depending on task requirements, a human is able to select distinct strategies such as the use of an ankle strategy and hip strategy to maintain their balance. Postural control actions often co-occur with other movements, and such movements may bring about a change from one type of postural coordination to another. The selection of a postural control strategy has typically been investigated by the transition of the center of mass (COM), center of pressure (COP), and in between angle joint motion along with their characteristics. In this paper, we proposed a method using the logistic function of the sigmoid model based on cross-correlation coefficient (CCF) data for investigating and observing the transition of postural control strategies of COM–COP and ankle-hip angles towards anterior–posterior (AP) continuous translation perturbation. Subjects were required to stand on the motion base platform where perturbations with an increasing frequency (0.2 Hz to 0.8 Hz) and decreasing frequency (0.8 Hz to 0.2 Hz) in steps of 0.02 Hz, were induced. As the frequency increased, the COM and COP displacements were decreased, with the opposite trend observable with decreasing frequency. This pattern was also observed at the head peak-to-peak amplitude. Meanwhile, ankle and hip angular displacements were increased during increasing frequency and decreased during decreasing frequency. In this paper, the proposed sigmoid model could identify the transition frequency of COM–COP and ankle–hip transition. The mean transition frequency of COM–COP during increasing frequency was 0.44 Hz, and the ankle–hip transition frequency was 0.42 Hz. Meanwhile, for decreasing frequency, the COM–COP transition frequency was 0.55 Hz, and for the ankle–hip transition the frequency was 0.56 Hz. With frequencies, both increasing and decreasing, the COM–COP and ankle–hip transition frequencies occurred almost at the same frequency. Furthermore, the transition occurred at a lower time scale during increasing frequency compared to decreasing frequency. In conclusion, the continuous translation surface perturbation provided information on the behavior of postural control strategies. A sudden change in 'phase angle' was observed, where either an ankle or hip strategy was implemented to maintain balance. Besides, the transition frequency of postural control strategies could be determined to occur between 0.4 Hz and 0.6 Hz, based on the average value, for healthy young subjects in the AP plane. Furthermore, the proposed sigmoid model was believed to be able to be used in the determination of transition frequency in postural control strategies.

**Keywords:** postural control strategies; sigmoid model; cross-correlation coefficient; continuous support surface translation perturbation; kinematics; kinetics; transition

## 1. Introduction

The ability to maintain balance in an upright position during quiet standing and dynamic task conditions is necessary for successful performance in daily life tasks [1,2]. Human postural control tends to initiate and constraint joint movement so that the center of mass (COM) is positioned over the base of support (BOS) and aligned with the center of pressure (COP), which is known as an equilibrium state. However, any external perturbation that is induced to those parts of the body will result in the shift of COM closer to the BOS border and interrupt alignment between COM and COP, which will cause a non-equilibrium state [3]. However, the balance control systems degenerate as we grow older, and this factor becomes the main reason for the high risk of falls among elderly people. Therefore, a better and deeper understanding of postural control strategies needs to be understood first before the development of training and rehabilitation tools.

Depending on task requirements, a human is able to select distinct strategies. Two primary postural control strategies were identified in upright bipeds, which are an ankle strategy and hip strategy [4,5]. The ankle strategy is viewed as an inverted pendulum with the motion around the ankle joint. Meanwhile, the hip strategy is used when the postural stability creates a particular constraint on the posture when the motion is around the hip joint. With the presence of a hip strategy, Buchanan and Horak suggested that a single link inverted pendulum will split into a multi-link model [6]. Besides these two primary postural control strategies, an additional strategy mode has also been identified that incorporates the ankle and hip strategies into a coordinated strategy. However, how these strategies are selected to maintain balance is still unknown and needs further investigation. In this study, we only observed the ankle and hip strategies since these are the primary strategies used in postural control.

Postural control actions often co-occur with other movements, and such movements may bring about a change from one type of postural coordination to another. This can be caused by the basic patterns of postural coordination being centrally represented by a set of motor programs, and postural transitions being behavioral consequences of changes between programs operating at the level of the central nervous system (CNS) [7,8]. These, as well as the changes between postural states, are consequences of the self-organized nature of the postural system, which exhibit properties of non-equilibrium phase transitions between attractors [9]. Typically, continuous relative phase (CRP) [10] or point estimate relative phase (PRP) [11] have been applied to investigate the coordination relation of two variables. The in-phase mode (~0°) indicates that two stationary signals move in the same direction, whereas the anti-phase mode (~180°) indicates that the signals are moving in the opposite direction. However, until now, there have been no studies that have mentioned the exact frequency at which the postural strategy changes happens. Other methods can also be considered in defining the coordination relation of two variables, such as cross correlation function (CCF) analysis. This analysis is a powerful, efficient, and relatively easy-to-apply tool for quantifying associations between variables. It also can be very useful for investigating spatial and temporal relationships between time-varying signals and can provide further insight into the coordination of movement, muscle activation patterns, and isolation noise within a signal. Furthermore, since the CCF coefficient data shows a resemblance to a sigmoidal pattern, we proposed that the transition frequency could be identified by using the logistic function of the sigmoid model. This is because the sigmoid model provides a good example of non-linear and quickly increasing functions of the probability of disclosure and makes computation easier [12]. While most traditional nonlinear activation functions are bounded, simpler piecewise linear activation functions have become popular because of their computational efficiency and robustness in preventing saturation [13]. Therefore, the sigmoid model is a reliable analysis technique, which mimics the physiologically based prediction of the input/output relation [14].

There have been several studies into the coordination of the body that affords stability in dynamic postural balance under both discrete [15,16] and continuous oscillation [17,18] of motion base support. Many researchers have reported the use of support surface perturbation as an experimental method to investigate different patterns of postural strategies [19–21]. According to previous studies, lower extremities play an important role in balance. The ascending sensory pathway from the sole then

regulates muscle activation to initiate ankle motion, then activating hip motion, and finally the upper body, including trunk and head [20]. Without sufficient activation or response from the lower extremities, it is difficult for nervous systems to provide feedback to generate an effective strategy and prevent falling. Therefore, it is believed that support surface perturbation, which acts as an external perturbation, is enough to induce postural control strategy patterns. Besides, this perturbation can replicate daily life activities such as slipping, tripping, and also stepping. Both increasing frequency and decreasing frequency were implemented in this paper. In previous studies, large step size changes of continuous translation frequency were used [10,21]. Motor sensory perception could sense slight changes of the translation frequency, which could lead to sudden changes in postural control strategies and the possibility of the subjects not reacting naturally. Therefore, a small step size of translation frequency was implemented in this paper to provide a more precise transition frequency.

This paper aims to investigate and observe the transition frequency of COM–COP and ankle–hip transition towards continuous translation frequency perturbation. The logistic function of the sigmoid model based on CCF coefficient data was used to determine the transition frequency. Besides, the effect of frequency order perturbation towards the transition frequency was also observed. Moreover, this paper is the first paper to introduce the sigmoid model based on CCF coefficient data for determining the transition frequency of postural control strategies. We examined the hypotheses that (1) a sudden shift from in-phase to anti-phase happens as effect of translation frequency perturbation; (2) the transition frequency range decreases with the implementation of a small step size in continuous translation perturbation; (3) the different order of translation frequencies affects the transition frequency; and (4) the sigmoid model based on CCF coefficient data can be used in determining the transition frequency of postural control strategies. The findings from this study will contribute to further understanding of human postural control strategies.

## 2. Methods

### 2.1. Participants

In this study, 20 healthy young male subjects participated (25.70 ± 4.42 years old, 65.85 ± 9.53 kg, 170.43 ± 6.63 cm). The subject sample size ($n$) was confirmed with sample size calculation. The minimum size for every parameter (increasing frequency (COM–COP, $n = 15$; ankle-hip, $n = 9$), and decreasing frequency (COM–COP, $n = 13$; ankle–hip, $n = 9$)) was obtained. From the calculation, it was confirmed that the current sample size was sufficient for conducting the experiment. Young and healthy subjects were selected for the determination of transition frequency due to their agility and ability for a better control strategy. All healthy subjects were free from neurological disorders, balance disorders, and vestibular function disorders. They had no history of neurological impairment. Information regarding the subjects' history of falls and physical condition were recorded as references. The experimental procedure was approved by the ethical committee of our research institute.

### 2.2. Experimental Procedures

Subjects were requested to stay standing quietly for 10 s to record quiet standing data. Then, subjects were exposed to external surface continuous translation perturbation with two types of frequency orders, which increased (0.2 Hz to 0.8 Hz) and decreased (0.8 Hz to 0.2 Hz) in 0.02 Hz steps with a displacement of 100 mm peak-to-peak. Buchanan and Horak stated that the transition frequency occurs at a frequency of 0.5 Hz and above [6]. Therefore, the selection of these frequencies, which is from 0.2 H to 0.8 Hz, with 0.5 Hz in between, was used in the determination of transition frequency. A 0.02 Hz step of frequency was implemented in order for subjects not to notice the slight changes of the translation frequency. Besides, this approach can avoid sudden changes, and the subjects can react naturally in executing postural control strategies. Each frequency changed after five oscillation cycles. The subjects were instructed to maintain their postural balance while the continuous translation perturbations were induced as they were barefoot, with eyes open, and focused at a mark that was set

at eye level. The total duration of the experiment was about 373 s for each subject in both increasing and decreasing frequencies. The subjects' vision and vestibular sense were not manipulated.

## 2.3. Experimental Apparatus

An external continuous translation perturbation in an anterior–posterior (AP) direction was produced by a 6-axis movable platform (MB-150, Cosmate, Tokyo, Japan). A force platform (9286A, Kistler, Yokohama, Japan) was used to derive the displacement of the body's COP and mounted on the moving platform. A 3-D motion capture analysis system with ten high-precision infrared cameras (Kestrel camera, Motion Analysis Corp., CA, USA) was used to record the motion of the passive marker attached over the joints of the subjects. Eighteen fixed reflective markers (placed at the 3rd metatarsal, lateral malleolus, lateral condyle, trochanter of the femur, iliac crest, acromion of spacula, top of the head, and four markers on the force plate) were attached over the subjects' joints. Both right and left knees were locked to prevent bias movement from the knees. This study focused on two segmental links of human postural strategies, which are ankle and hip strategies. Therefore, the absence from the knee joint was necessary to purely obtain the ankle and hip joint data. The method used to lock the knee joint was by using a pair of wood splints sandwiched around the knee joints in account of the subject comfort.

## 2.4. Data Collection

Motion capture data, which are marker data, underwent their own post-processing in order to create and gather the marker names and coordinates. All disconnected frames of marker positions needed to be corrected and smoothed to eliminate noise. Furthermore, force plate data were also filtered with a 2nd Butterworth filter with a cut off of 6 Hz to eliminate noise, especially from the power line and movement. Each data point was then resampled to a sampling frequency of 200 Hz. The coordinated data from motion capture were collected as raw data for the calculation process. The COM was calculated from the eight-segment model. The total body COM position was obtained by the weighted summation of the individual segment COM position, as mentioned in Equation (1). Constant value for each segment was shown as in Table 1 [22]. The ground reaction force ($F_v$) was obtained by summing up all the vertical forces from the strange gauge (i.e., $f_{z1}$, $f_{z2}$, $f_{z3}$, and $f_{z4}$) of the force plate, as shown in Figure 1. The force plate moment in the x-axis ($M_x$) was generated by these vertical forces, as shown in Equation (2). The joint movement coordinates ($x$, $y$, $z$) obtained from motion analysis systems with a 1 kHz sample rate were used to measure joint angle displacement ($\theta_{ankle}$ and $\theta_{hip}$) and body segment length ($h_{ankle}$, $h_{hip}$, and $h_{seg}$) for segmental COM location. The COP displacement was determined from Equation (3) below where $d_z$ was the distance from the surface to the platform origin. The ankle and hip angle displacement were calculated with Equation (4) by using 2 vectors, as shown in Figure 2. The average of head peak-to-peak amplitude for each frequency was also calculated.

**Table 1.** Constant value for each segment [22].

| Segment | Segment Weight/Total Body Weight |
|---|---|
| Head, arm, trunk (HAT) | 0.536 |
| Pelvis | 0.142 |
| Left/Right Thigh | 0.1 |
| Left/Right Leg | 0.0465 |
| Left/Right Foot | 0.0145 |

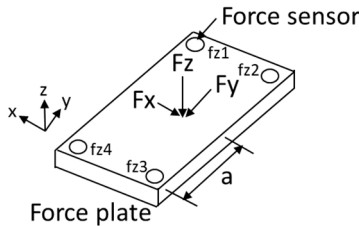

**Figure 1.** Force distribution on force plate.

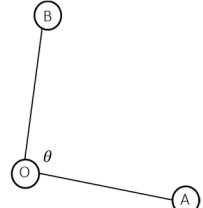

**Figure 2.** Joint angle.

The COM and COP displacement in the AP direction was as follows:

$$COM_{AP}[mm] = \sum \left(COM_{each\ seg} \times constant\ value_{each\ seg}\right) \tag{1}$$

where the constant value is in the table below.

$$M_x = a(f_{z1} + f_{z2} + f_{z3} + f_{z4}) \tag{2}$$

$$COP_{AP}[mm] = \frac{M_x - \left(F_y{\cdot}d_z\right)}{F_v} \tag{3}$$

where *a* is the sensor offset value and $F_y$ is the horizontal component in *y*-direction force.

The ankle and hip angle displacement in the AP direction were as follows:

$$\cos\theta(deg) = \frac{\vec{A}{\cdot}\vec{B}}{|A||B|} \tag{4}$$

*2.5. Data Analysis*

2.5.1. Cross-Correlation Coefficient

This study focused on observing the changes of postural control strategies, especially regarding the transition frequency of COM–COP and ankle–hip angle toward increasing and decreasing frequency translation perturbation. In this paper, CCF analysis was used to observe the correlation between COM–COP and ankle–hip angle. This analysis is a powerful, efficient, and relatively easy-to-apply tool for quantifying associations between variables. It also can be very useful for investigating spatial and temporal relationships between time-varying signals and can provide further insight into the coordination of movement, muscle activation patterns, and isolation noise within a signal [23]. The coefficient value range was between −1.0 and 1.0. A CCF coefficient of −1.0 shows a perfect negative correlation, while a coefficient of 1.0 shows a perfect positive correlation. A calculated value greater than 1.0 or less than −1.0 means that there was an error in the experimental, data collection,

and processing measurement. Cross-correlation involves correlating two different time-varying signals against each other. It was calculated using Equation (5), where $x$ and $y$ are the sample data:

$$r = \frac{n(\sum xy) - (\sum x)(\sum y)}{\sqrt{\left[n\sum x^2 - (\sum x)^2\right]\left[n\sum y^2 - (\sum y)^2\right]}} \tag{5}$$

There was a total of 31 frequencies, from 0.2 Hz to 0.8 Hz and 0.8 Hz to 0.2 Hz with 0.02 Hz steps. Each frequency comprised 5 cycles, which gave a total of 155 cycles except 0.2 Hz for increasing frequency and 0.8 Hz for decreasing frequency translation perturbation, which comprised 6 cycles. The first cycle of 0.2 Hz and 0.8 Hz was eliminated in order to cut out unnecessary movement at the beginning of the experiment. An average of 5 cycles from every frequency was calculated. Then, by using the value of average cycle, CCF coefficients were calculated for COM–COP and ankle–hip angle in order to observe the correlation of these two cycles.

2.5.2. Logistic Function (Sigmoid Model)

The sigmoid model provides a good example of non-linear and quickly increasing and decreasing functions of the probability of disclosure and makes computation easier [12]. Besides, the sigmoid model is a reliable analysis technique that mimics the physiologically based prediction. In this paper, the CCF coefficient data distribution graph exhibits a sigmoidal pattern at increasing and decreasing stimulation intensities. Therefore, we proposed the sigmoid model in order to determine the transition phase of human postural control strategies. A general least square model similar to one developed in the investigation of the ascending limb of all recruitment curves and transcranial magnetic stimulation (TMS) research was used [13]. In this paper, a custom four-parameter sigmoid model was modified as shown in Equation (6) below:

$$F_x = \frac{L}{1 + e^{k(x-x_0)}} + y_0 \tag{6}$$

where $L$ is the vertical range of model, $k$ is the gradient or slope of the model, and $x$ is the CCF coefficient data distribution. Furthermore, $x_0$ is determined as the transition frequency, and $y_0$ is the vertical correcting position of the model.

The sigmoid model provides a good example of a non-linear model, can predict the probability as an output, and makes computation easier. Therefore, the sigmoid model is reliable, which mimics the physiologically based data as in this study is from CCF coefficient data distribution [14]. This method utilizes curve fitting using a logistic function, which is optimized using the particle swamp optimization (PSO) technique. With this technique, the model with the lowest error was chosen as the best model providing the sigmoid model. In this paper, the lower frequency was referring to the frequencies below 0.5 Hz and high frequency was for frequencies above 0.5 Hz.

This model clearly can divide upper and lower distributions and the slope in between the distribution, as shown in Figure 3. In order to find the transition frequency for each subject, first we determined the initial slip and convergence of the model by the velocity of the sigmoid model data. The slope between the initial slip and convergence was defined as a transition phase, indicating that transition happens from the positive phase shift to the negative phase shift and vice versa. We calculated the coefficient of determination $\left(r^2\right)$, which is also defined as global goodness-of-fit of the sigmoid model, to clarify the fitness of the model towards CCF coefficient data. $r^2$ was calculated in order to define the fitness of the model since the model is a linear regression. The $r^2$ value of more than 0.039 with one variable was indicated as acceptable to the fitness of the model.

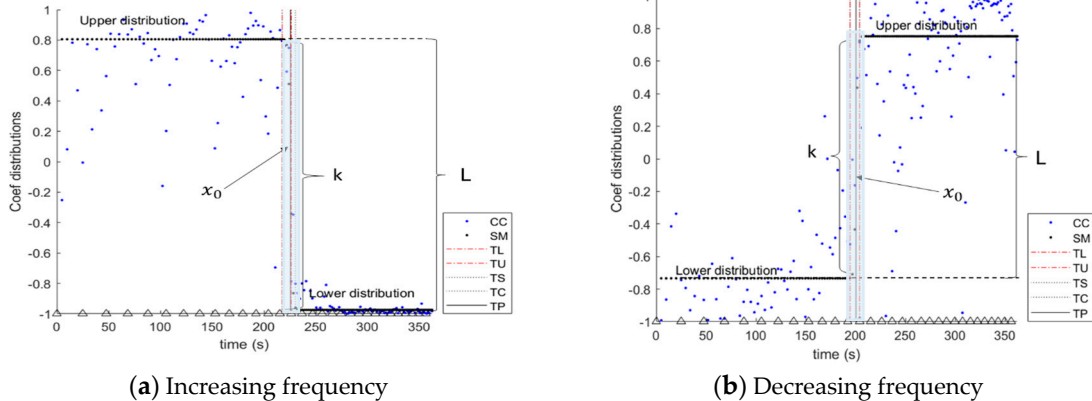

(**a**) Increasing frequency     (**b**) Decreasing frequency

**Figure 3.** Example of cross-correlation function (CCF) data distribution with the sigmoid model graph at (**a**) increasing and (**b**) decreasing frequency order conditions. The area between the upper distribution and lower distribution of each condition was determined as the transition phase (indicated as a light blue area) where the transition frequency was observed. CC: Cross-correlation function coefficient data; SM: sigmoid model; TL: transition lower; TU: transition upper; TS: transition slip; TC: transition convergence; TP: transition frequency point.

### 2.6. Statistical Analysis

The transition frequencies of all subjects are summarized in results section in terms of frequency. A comparison of the transition frequency between increasing and decreasing frequencies of COM–COP and ankle–hip angles was analyzed by using paired *t*-test analysis with a significance level of $p < 0.05$. The same statistical test was used to analyze a comparison between COM–COP and ankle–hip transition frequency in increased and decreased frequencies. Furthermore, two-way ANOVA analysis was used to observe the comparison between individual subjects. The Bonferroni post hoc test was used to determine the differences in all levels of comparison for the independent variables. All statistical analyses were completed using the MATLAB software.

## 3. Results

### 3.1. Displacement of Kinematics Parameters

All subjects were able to accomplish the given tasks perfectly. Figure 4 shows the waveforms of the representative subjects for COM, COP, ankle, and hip angle displacement during increasing and decreasing frequencies. Based on these waveforms, the COM and COP displacements decreased in frequency during increasing frequency. For ankle and hip angular displacement, the displacements were increased. Meanwhile, COM and COP displacements were increased, and ankle and hip angular displacements were decreased during decreasing frequency. With the presence of the transition between COM–COP and ankle–hip, at a certain time, the phase was changed from in-phase to anti-phase.

Figure 5 illustrates the average value of head peak-to-peak amplitude for each frequency during increasing and decreasing frequency. During increasing frequency perturbation, the peak-to-peak amplitude was decreasing. It was observed that the head peak-to-peak amplitude dropped about 50% between 0.42 Hz and 0.46 Hz. Besides, during decreasing frequency perturbation, the head peak-to-peak amplitude was gradually increasing. However, during low frequency, the head peak-to-peak amplitude during decreasing frequency was lower than increasing frequency.

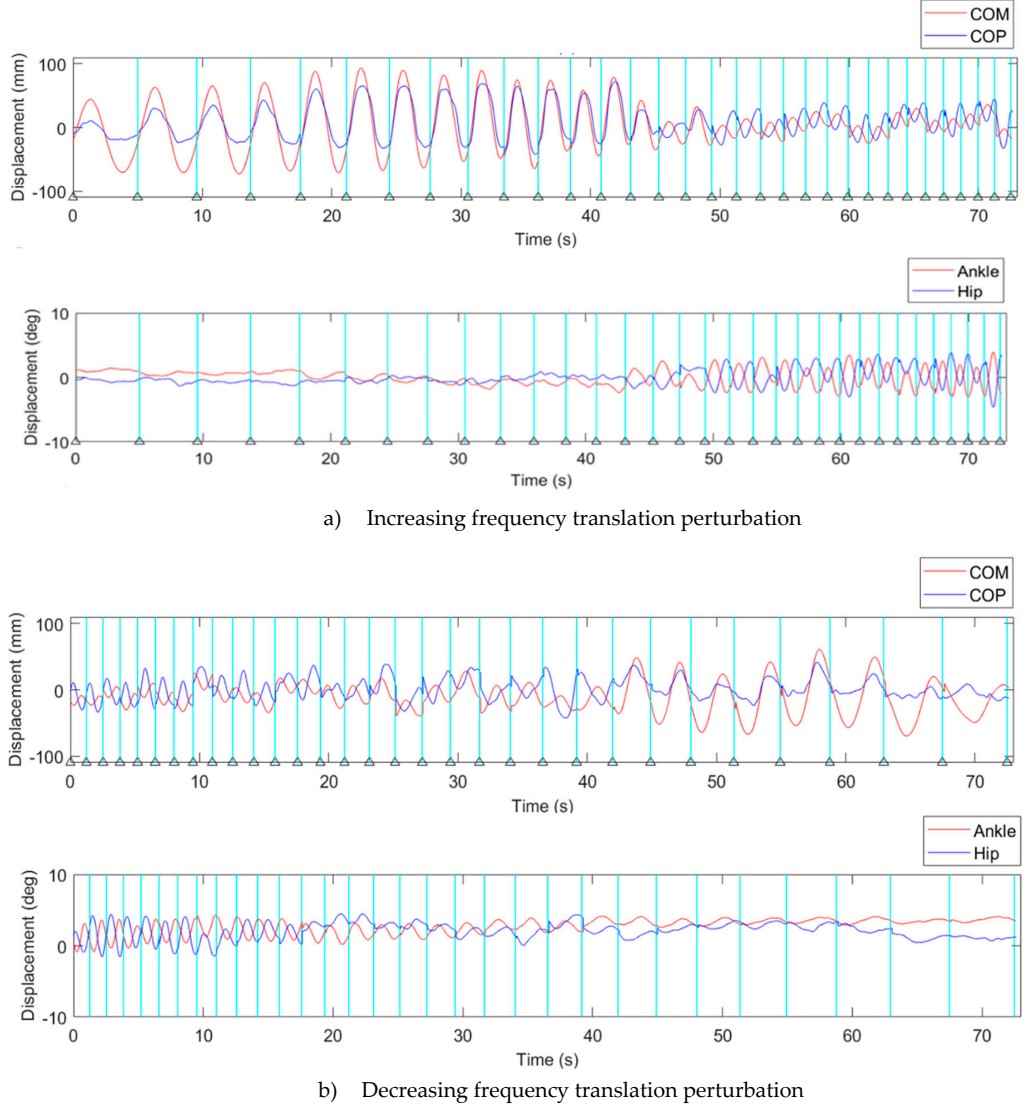

**Figure 4.** The representative waveform of the center of mass (COM), center of pressure (COP), ankle joint, and hip joint displacement based on time during (**a**) increasing frequency and (**b**) decreasing frequency translation perturbation.

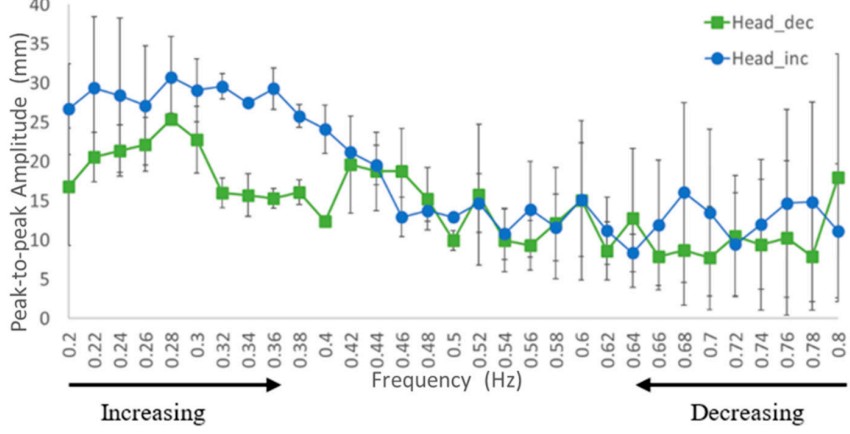

**Figure 5.** The average value of head peak-to-peak amplitude for each frequency during increasing and decreasing frequency.

### 3.2. Transition Frequency

The relation of COM–COP and ankle–hip angle was analyzed by using CCF coefficient data, and the transition frequency was determined with the sigmoid model. The COM–COP and ankle–hip sigmoid models of the representative subject are shown in Figure 6. All subjects showed the changes from upper to lower distribution and vice versa of CCF coefficient data, which can be explained as the degree of freedom of human balance being changed from one degree of freedom (DOF) to two DOF. The COM–COP and ankle–hip transition frequency for every subject and perturbation conditions are summarized and shown in Figure 7. Even though all subjects were free from neurological disease, they showed different transition frequencies at both COM–COP and ankle–hip transitions.

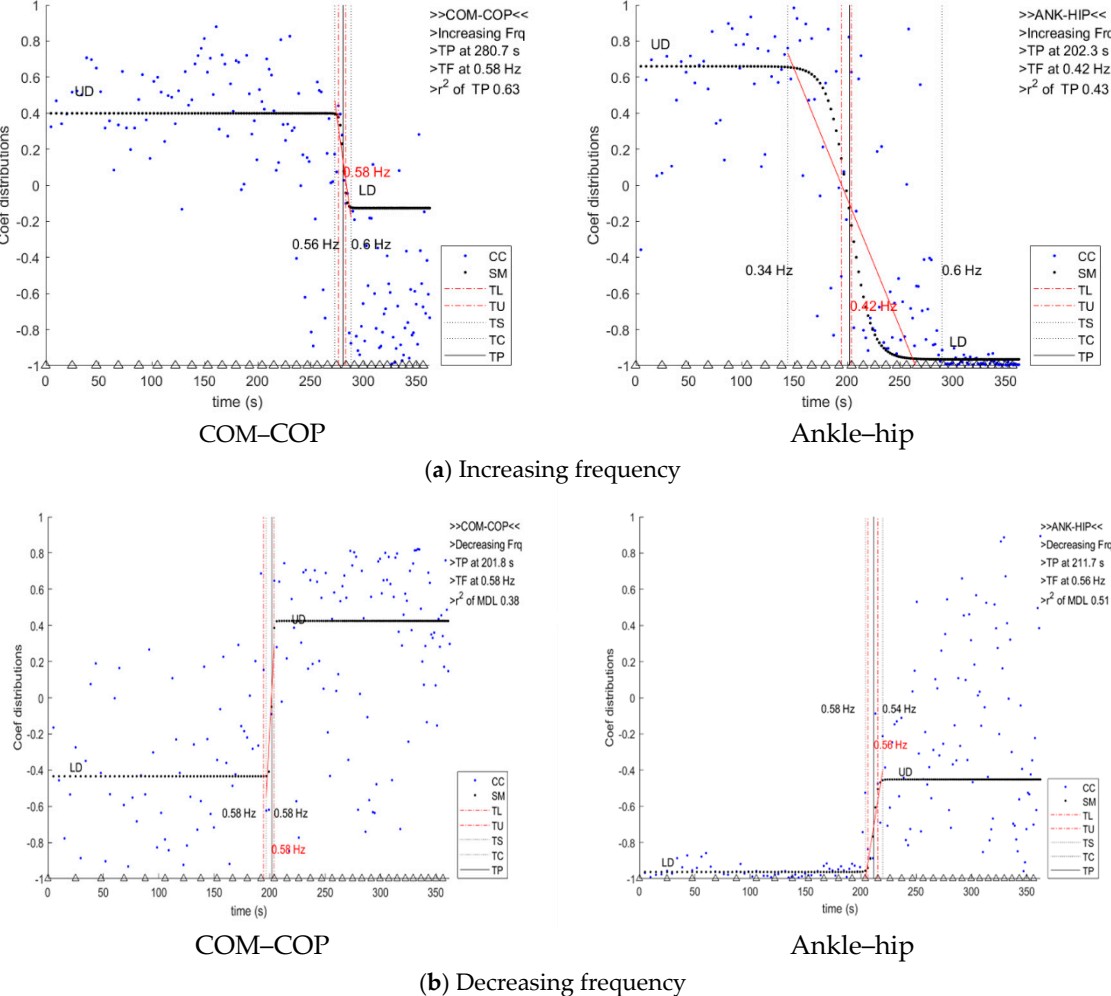

(**a**) Increasing frequency

(**b**) Decreasing frequency

**Figure 6.** The representative of CCF coefficient data distribution with the sigmoid model during (**a**) increasing frequency and (**b**) decreasing frequency translation perturbation condition. Each blue point represents a CCF coefficient data for each frequency. The black line represents the sigmoid model. UD: Upper distribution; LD: Lower distribution.

For the COM–COP transition, the transition frequency varied between subjects. From the total of 40 transitions frequencies of COM–COP and ankle–hip for the 20 subjects, 26 transitions frequencies showed a transition of both COM–COP and ankle–hip transition at lower frequencies during increasing frequency perturbation conditions. For decreasing frequency, 16 transition frequencies showed the transition in the lower frequencies. The mean transition frequency of COM–COP was 0.44 Hz ($\pm$0.12) during increasing frequency, while it was 0.55 Hz ($\pm$0.12) during decreasing frequency. Furthermore, the mean transition frequency of ankle–hip during increasing frequency was 0.42 Hz ($\pm$0.12) and

0.56 Hz (±0.12) during decreasing frequency. From the mean value of transition frequency, it was observed that the COM–COP transition and ankle–hip transition occurred almost at the same frequency during both increasing and decreasing frequency. A significant difference was observed between increasing and decreasing frequency for COM–COP ($p = 0.003$) and ankle–hip transition frequencies ($p = 0.008$). However, no significant difference was observed between COM–COP and ankle–hip transitions frequency in both frequency conditions (increasing ($p = 0.56$, ns); decreasing ($p = 0.17$, ns)). No significant difference was observed between subjects during both frequency order conditions (increasing ($p = 0.08$, ns) and decreasing ($p = 0.34$, ns)). A significant difference was observed between increasing and decreasing frequency perturbation for head peak-to-peak amplitude ($p = 0.0361$). All subjects showed $r^2$ value more than 0.039, indicating that the model significantly fit with the CCF coefficient data distribution.

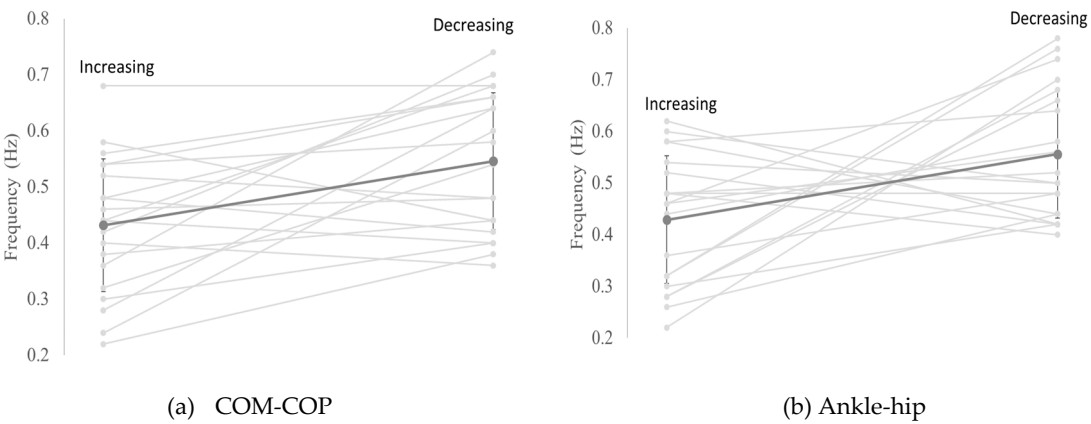

(a)   COM-COP

(b) Ankle-hip

**Figure 7.** The transition frequency of (a) COM–COP and (b) ankle–hip during increasing and decreasing frequency translation perturbation. The dark line represents the mean value of transition frequency.

## 4. Discussion

The primary aim of this study was to observe and investigate the transition frequency of postural control strategies by using a proposed sigmoid model based on CCF coefficient data distribution. This paper is the first to use a sigmoid function model in order to observe the postural strategy transition frequency. The finding shows that the subjects' COM and COP displacements were decreased during increasing frequency and increased during decreasing frequency. In this paper, the transition frequency could be obtained from the proposed sigmoid model based on CCF coefficient data. All subjects showed difference transition frequencies at both COM–COP and ankle–hip. The transition frequency of ankle–hip occurred earlier than that of COM–COP during both increasing and decreasing frequencies.

### 4.1. Sigmoid Model Based on Cross-Correlation Coefficient Data for the Determination of Transition Frequency

Postural strategy transition can be defined as the changes of postural states between two attractors that result from the consequence of changes between programs operating at the level of the central nervous system (CNS) or of the self-organized nature of the postural system [9]. To date, several studies have investigated the transition between postural strategies by using a relative phase, either using point estimate relative phase (PRP) or continuous relative phase (CRP) [10,24–26]. However, in the present study, we proposed an analysis of the transition of postural strategy coordination by using a sigmoid model based on CCF coefficient data. The CCF coefficient is a statistical measure that calculates the strength of the relationship between the relative movements of two variables. Besides, since the CCF coefficient data distribution exhibits a similar pattern to the sigmoidal pattern, the sigmoid model was proposed to determine the transition frequency of the postural control strategies. This proposed model was shown to be suitable with any CCF coefficient data distribution even though the data were spread in a large distribution.

### 4.2. Kinematic Characteristic of Continuous Translation Perturbation Frequencies

The different frequencies, increasing and decreasing continuous translation perturbation platform, provided an experimental manipulation to investigate postural control strategies [10]. Based on kinematic results, COM and COP displacements were decreased towards increasing frequency and increased during decreasing frequency. This is in contrast with ankle and hip angle, where angular displacement increased during increasing frequency and decreased during decreasing frequency. Sensory information from visual, vestibular, and somatosensory systems is regulated by the central nervous system (CNS) to maintain balance and postural orientation. Different frequencies exposed to the body will change the coordinative patterns of the head, trunk, and legs to accommodate the different action on the body, such as the transition of postural strategy from ankle strategy to hip strategy [17]. Translating the body at different frequencies also moves the sensory systems within and outside of their optimal operating ranges. In this paper, at a low frequency, the subjects appeared to "ride" the platform depending on the movement of the platform itself. As suggested by Buchanan and Horak, the COM amplitude decreased during increasing frequency and the head motion became fixed in space, allowing vision scene oscillation produced by the translating support surface [6]. At the high frequency, the CNS tended to apply accurate control to reduce kinematic displacement in order to maintain balance in the desired position [27]. Besides, the CNS may suppress anticipatory postural adjustments (APAs) since APAs can reduce the effect of the forthcoming body perturbation [28]. Focusing on COP displacement, at a low frequency the displacement was smaller than COM and head displacement. In spite of this, the COP displacement was higher than COM at a high frequency, which agreed with the result of David A. Winter [2]. This indicated that the COP was greater in order to keep the COM within the COP line to maintain equilibrium. As shown in the results, at a low frequency the COM and COP were in-phase. As the frequency increased, COM and COP phases became anti-phase. An increasing transition frequency causes the COP amplitude to decrease and disturb the postural strategy of a subject as the COM amplitude can be out of bound, where a good prosecution of posture control strategy always requires inbound COM amplitude to avoid a stepping strategy. To avoid the involvement of a stepping strategy, the body increases joint stiffness and moment at the ankle joint [29]. At the lower frequency, the ankle strategy was used as a balance strategy. However, the physiological limits of performing the ankle strategy will be reached during increments in translation frequency; at this instance, it will try to compensate the balance strategy with contribution from hip movement [30,31]. This action of hip movement changes the COM acceleration, which in fact will affect the postural control strategy from an ankle to hip strategy. This changes the transition phase from in-phase to anti-phase while adapting to the perturbation translation's acceleration [32]. However, based on the result, the ankle–hip transition usually comes earlier than that of the COM–COP transition.

### 4.3. From a Single-Linked Model to Multi-Segmental Model

Quiet standing tends to approximate a single-link inverted pendulum. Recent studies have pointed out that hip joints play an important role in maintaining balance even in quiet standing [20,33–36], as well as in dynamic tasks, which show more ankle–hip joints characteristics [36] and which we can consider as a multi-segmental model. In this paper, the sigmoid model has shown that there was an upper and lower distribution of CCF coefficient data of COM–COP and ankle–hip, which revealed that there was a sudden transition from low to high frequency and vice versa at a certain frequency. These pattern changes are consistent with the results from Ko et al. [24]. The joint degree of freedom was regulated by a small amplitude as the motion from the moving platform was at a low frequency. However, as the motion was increased, the joint degree of freedom motion showed a large amplitude and forced the postural strategy to re-organize and dissipate the large force; i.e., the head fixed pattern and transition from ankle to hip strategy and vice versa.

The means of transition frequency for COM–COP and ankle–hip transitions were 0.44 Hz and 0.42 Hz for increasing frequency and 0.55 Hz and 0.56 Hz for decreasing frequency, respectively. This finding was parallel with Aviroop and Karl, who obtained a coordination change around of

0.4 Hz–0.6 Hz for the COM–COP transition frequency [11]. The transitions frequencies in both increasing and decreasing frequency were almost in the same frequency for both COM–COP and ankle–hip. Even though the transition occurred almost at the same frequency, a slightly difference was observed where the ankle–hip transition frequency was earlier than that of COM–COP. This finding was consistent with Ko et al., who stated that the COM–COP coordination is relatively robust and on a slower time scale of change compared to the coordination of joint components and their individual motions [24]. As the perturbation applied originated from the lower extremities, the perturbation moved to the sole then regulated muscle activation, which increased joint stiffness and moment at the ankle joint. Then, as the physiological limit was reached, the hip movement was involved. Within this process, the COM and COP positions were also moved from the origin.

The transition happened when the initial pattern was undergoing a state of instability. The differences of transition frequency were observed between increasing and decreasing frequency, where the mean transition frequency for increasing frequency condition occurred at the lower frequency than decreasing frequency condition. These different values for the transitions indicated that the coordination mode observed is dependent upon the direction of the changes in the control parameter (platform frequency) [37,38]. These results exhibit a hysteresis phenomenon when the frequency transition value is varied at different conditions of perturbation frequency. Hysteresis in this study means that the tendency for the postural control systems itself to remain in the current behavior state as the control parameter moves through the transition region, provides different frequency transition values depending on the direction of the control parameter [39]. Furthermore, the coordination of the postural control strategy was more stable when moving from anti-phase to in-phase (decreasing frequency) compared to in-phase to anti-phase (increasing frequency) [11]. In decreasing frequency, the instabilities decreased as the frequency perturbation decreased, which resulted in the earlier transition frequency compared to increasing frequency. Therefore, the mean value of decreasing transition frequency occurred earlier than that of increasing transition frequency, which indicates the subjects easily coordinated the postural strategies during high frequency perturbation tasks.

The differences and variability of the transition point frequency of COM–COP and ankle–hip among the subjects were also observed. The variation of transition frequency among the subjects might be related to the balance ability of each subject [40]. It was suggested that patients with low scores of the balance ability test, i.e., Functional Reach Test, have high joint stiffness [30]. However, in this study, no correlation was observed between balance ability and transition frequency among the subjects. This may have happened because there is no specified condition implemented that can normalize posture control strategies.

### 4.4. Study Limitations

Two limitations to the present study should be noted. The first is that only healthy and young subjects participated with high agility and flexibility, as the capability of each subject in terms of their motor sensory and reflex were considered. Without these attributes, the possibility of the subject being unable to produce a good posture control strategy during the manipulation of translation perturbation is high. In the future, we will include a contribution from subjects with impaired conditions such as having bad vision or vestibular disorder to determine this factor's influence in the identification of transition frequency. Besides for that, elderly people also need to be included. The second limitation is that both knees of the subjects were locked during perturbation to minimize the knee involvement in the postural control strategy, as our focus was to investigate the transition of ankle and hip strategy. This is because during an upright bipedal state, the major contribution of posture control strategies comes from ankle and hip movements, while the knee is involved only when the subject is out of balance [41].

## 5. Conclusions

In conclusion, a transition frequency identification of postural control from an ankle to hip strategy was presented. Two types of data were used: the first was COM–COP displacements and the second was ankle–hip angular displacements. These data were analyzed to obtain CCF coefficient data under increasing and decreasing continuous translation surface perturbation at different frequencies. The transition point of postural control strategies was identified based on the transition phase of CCF data (in-phase to anti-phase). This was achieved by implementing the logistic function of the sigmoid model to differentiate the upper and lower distribution of CCF coefficient data. Two tests were performed to determine the significant difference between frequencies and types of data; the first was a paired *t*-test and second was a two-way ANOVA. It was observed that ankle–hip angle opposed the COM–COP displacements during increasing and decreasing perturbation frequency, where the COM and COP displacements were decreased during increasing frequency perturbation while the ankle and hip angle displacement were increased and vice versa for decreasing frequency perturbation. Besides, the peak-to-peak amplitude of head displacement was decreased as the perturbation was increasing and vice versa. At some instant, there is a sudden change in the 'phase angle' of the data, where either the ankle strategy becomes passive or the hip becomes active. Based on the results, the mean transition frequency of COM–COP and ankle–hip for all subjects was determined. It showed the transition of COM–COP was evident at 0.44 Hz and the transition of ankle–hip was at 0.42 Hz during increasing perturbation. Meanwhile, a transition frequency of 0.55 Hz (COM–COP) and 0.56 Hz (ankle–hip) was observed during decreasing perturbation. Therefore, it was concluded that the transition frequency of postural control strategies for both the COM–COP and ankle–hip data of healthy subjects was apparent between 0.4 Hz and 0.6 Hz. The transition frequency was varied according to the direction of the platform frequency. Besides, the variation in the transition frequency of each subject occurs because there is no specified condition implemented that can normalize the posture control strategy such as a light touch and body-harness support. In addition, the implementation of a small step size of 0.02 Hz during increasing and decreasing continuous translation surface perturbation gives a more accurate reading of transition frequency as compared with previous studies, which indicated that the range of transition happened at 0.5 Hz and above. This study provides a new approach for analyzing the transition phase of postural control strategies.

**Author Contributions:** Conceptualization, M.A.M.D.; data curation, N.F.F.M.R.; formal analysis, N.F.F.M.R.; funding acquisition, S.-I.Y.; investigation, N.F.F.M.R.; methodology, N.F.F.M.R.; software, M.A.M.D.; supervision, S.-I.Y.; validation, N.F.F.M.R.; writing—original draft, N.F.F.M.R.; writing—review and editing, S.-I.Y.

**Funding:** This research received no external funding.

**Conflicts of Interest:** The authors declare no conflict of interest.

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
