# Peer review of "Estimation of Transition Frequency during Continuous Translation Surface Perturbation"

_applsci, doi:10.3390/app9224891_

Round 1
Reviewer 1 Report
Thank you for submitting this paper to Applied Sciences. The manuscript under consideration: " Estimation of Transition Frequency during Continuous Translation Surface Perturbation " is an interesting article on an important topic in Applied Sciences. Some revisions are listed below.
1. Sample size reporting is imperfect. Were power calculations performed a priori? Please provide all parameters for the sample size calculation in the Methods.
2. "A comparison of the transition frequency between increasing and decreasing frequencies of COM-COP and ankle–hip angles and a comparison between COM-COP and ankle-hip transition frequency in increased and decreased frequencies were analyzed by using paired t-test analysis with a significance level of p<0.05." Statistical reporting is imperfect. Were the data statistically tested for normal distribution?
Reviewer 2 Report
Review Report – ApplSci - 620013
A brief summary
This study has ambition to contribute to further understanding of human postural control strategies, determined by sigmoid model based on CCF coefficient data. It observes the transition frequency of COM-COP and ankle-hip transition towards continuous translation frequency perturbation.
Broad comments
Introducing overview was mostly supported by methodology used to fulfil aim of the experiment. Experimental design was generally appropriate with adequately presented methods, followed and supported by relevant and precise conclusions and explanation of limitations of the study. However, second limitation, i.e. postural control strategies in relation to methodology of locking both knees should be presented and supported by previous research and experimentally more thoroughly and precisely. Otherwise, it may, as a major flaw, bring potential suspicions in validity and exactness of findings and results.
Specific comments In lines 123-126 “sample size (n) was confirmed with sample size calculation” – what was effect size, and what exact calculation was used? In line 156 – “Both right and left knees were locked to prevent bias movement from the knees”. – Even if it was regular in previous research, it needs practice to exclude influence of locking to hip or ankle strategies. In other words – locking generated additional different adaptations in hip and ankle strategies. Explain methodology of ‘locking’ in more details. In lines 205-6: “number greater than 1.0 or less than -1.0 means that there was an error in the measurement” is not absolutely true. It is often an error in experimental design or sometimes irresponsible data collection, processing, analysis or else. In lines 233-4: “The sigmoid model is a clinically or phenomenological based model which performs based on the observation of postural behaviour through CCF coefficient data distribution.” Please rephrase and reinterpret this statement in more accurate and discrete way. In Lines 419-424: “second limitation is that both knees of the subjects were locked during perturbation to minimize the knee involvement in postural control strategy, as our focus was to investigate the transition of ankle and hip strategy. This is because, during an upright bipedal state, the major contribution of posture control strategies comes from ankle and hip movements, while the knee is involved only when the subject is out of balance” – this statement should be supported by previous research and experimentally (or at least by one of those), especially since it can produce false positive relations mostly due to heterogeneity in morphology of subjects, not only in ankle-hip strategies, perturbations or else. If omitted, this moment may represent major flaw to this (very nice and interesting) research. All references should be presented in standardized way. Top of Form.
Round 2
Reviewer 2 Report
Author’s response and content of reply sufficiently supported their arguments, therefore accepted.
Author Response
The action has been taken as suggested.

This manuscript is a resubmission of an earlier submission. The following is a list of the peer review reports and author responses from that submission.
Round 1
Reviewer 1 Report
Comments to the Author
I am reviewing the article “Estimation of Transition Phase during Continuous Translation Surface Perturbation using Sigmoid Function based on Cross-correlation Coefficient Data”. The article provided the estimation of transition phase during sigmoid function and continuous support surface translation perturbation. The sample number is few, and the paper had some problem that is misleading the results. Here are some comments on the article:
1. The major conclusions in this study is that the proposed method of sigmoid model based on CCF coefficient data is can be used in determination of transition point of postural coordination, but in the study only 11 young healthy participated. How did the authors determine the sample appropriate size?
2. Please identify how to recruit participants.
3. Why did authors select male only? From the male and female, it is possible to obtain measurements of postural coordination on difference in sex. Authors should clearly describe the purpose of selecting male only.
4. Authors did not describe the process of barefoot for 10s to record quiet standing data. Is this the most optimal number of seconds to answer the research question?
5. What is the physiological basis for the selected frequencies?
6. How the center of pressure data and Motion capture is processed? What was the sampling frequency and was any filter used?
7. Is it indicated that the normal distribution for each variable is confirmed using the Shapiro Wilk test?
8. Additional correction tools (e.g., Bonferroni correction) should be utilized to confirm the results of Two-way ANOVA analysis.
9. What is the physiological basis for the selected frequencies? Discussion is neurophysiological explanation is inadequate.
10. What are the specific effects of the different frequencies on COM-COP?
11. What is the clinical interpretation of the sigmoid function model based on CCF coefficient technique?
Author Response
Point 1: The major conclusions in this study is that the proposed method of sigmoid model based on CCF coefficient data is can be used in determination of transition point of postural coordination, but in the study only 11 young healthy participated. How did the authors determine the sample appropriate size?
Response 1: Based on the previous studies, eleven healthy subjects are regarded sufficient which can provide difference pattern of cross-correlation distribution.
Point 2: Please identify how to recruit participants.
Response 2: A young and healthy subjects were selected for determination of transition point because of their agility and ability for better control strategy coordination. (Line 118-120)
Point 3: Why did authors select male only? From the male and female, it is possible to obtain measurements of postural coordination on difference in sex. Authors should clearly describe the purpose of selecting male only.
Response 3: This paper scope not focusing on different gender. This paper focusing on the methodology of determination of transition point.
Point 4: Authors did not describe the process of barefoot for 10s to record quiet standing data. Is this the most optimal number of seconds to answer the research question?
Response 4: Based on previous studies, the range of time taken for quiet standing was in between 10-30 seconds and that is not crucial for data analysis. Therefore, we can clarify that 10 seconds is sufficient for data analysis.
Point 5: What is the physiological basis for the selected frequencies?
Response 5: Buchanan has stated that transition point has occurred at the frequency more than 0.5Hz. Therefore, the selection of these frequencies which is from 0.2H and 0.8Hz; with 0.5Hz in between; was used in the determination of transition point. The selected frequencies of 0.02Hz were for subjects to unnoticed the changes of the frequency and to avoid sudden changes in postural strategy coordination. (Line 128-132)
Reference
Buchanan, J. J.; and Horak F.B. Emergence of Postural Patterns as a Function of Vision and Translation Frequency. Journal of Neurophysiology 1999, 81, pp. 2325–39. https://doi.org/Pdf/Buchanan&Horak_1999.pdf.
Point 6: How the center of pressure data and Motion capture is processed? What was the sampling frequency and was any filter used?
Response 6: Have been stated in the paper at line 152 but some additional information was added. Motion capture data which is data marker data undergo its own post-processing in order to create and gather marker’s name and coordinate. All disconnected frame of marker position needs to be corrected and smoothed to eliminate noise. Furthermore, force plate data were also filtered with 2nd Butterworth filter with a cut off 6Hz to eliminate noise especially from the power line and movement. Each data was then resampled to sampling frequency of 200Hz. the coordinated data from motion capture were collected as raw data for calculation process. (Line 150-156)
Point 7: Is it indicated that the normal distribution for each variable is confirmed using the Shapiro Wilk test?
Response 7: The data in this paper is not a scattered data/random data where the data were based on frequency/time domain which is steadily increased and decreased. Therefore, the normal distribution of data is not needed to be confirmed.
Point 8: Additional correction tools (e.g., Bonferroni correction) should be utilized to confirm the results of Two-way ANOVA analysis.
Response 8: This analysis gave no effect in the transition phase of postural strategy. Therefore, we decide to remove this data from the paper.
Point 9: What is the physiological basis for the selected frequencies? Discussion in neurophysiological explanation is inadequate.
Response 9: The different scaling of frequencies in increasing and decreasing of platform frequency provided an experimental manipulation to investigate postural control strategy coordination under different parameter region. Sensory information from visual, vestibular and somatosensory systems is regulated by central nervous systems (CNS) for maintaining balance and postural orientation. Different frequencies exposed to the body will change the coordinative patterns of the head, trunk, and legs to accommodate the different acting on the body such as the transition of postural strategy from ankle strategy to hip strategy. Translating the body at different frequencies also moves the sensory systems within and outside of their optimal operating ranges. (Line 304-310)
Point 10: What are the specific effects of the different frequencies on COM-COP?
Response 10: As shown in results, at the low frequency the COM and COP were in-phase and as the frequency increased COM and COP phase become anti-phase. Increasing the frequency will give effect to the increasing COP amplitude which can endanger balance and at the same time the moment at the ankle also increased until bordering the physiological limits of the systems. At this moment, the systems switched from in-phase to anti-phase which resulted in changes of strategy to hip strategy to produce a given magnitude of horizontal acceleration of COM. Thus, any external or internal perturbation that pushed the COM to the limits of base of support and changed the alignment between COM and COP may lead to the postural control changes. (Line 322-329)
Point 11: What is the clinical interpretation of the sigmoid function model based on CCF coefficient technique?
Response 11: The sigmoid function model is a phenomenological based model which is performed based on observation of postural behaviour through CCF data. This method utilizes a curve fitting using logistic function; which is optimized using particle swamp optimization method. Thus, the sigmoid model pattern may be a new reliable method in investigating the transition of postural strategy coordination pattern.

Reviewer 2 Report
Review Report – Applied Sciences - 539610
• A brief summary
a new method by using logistic function (sigmoid) based on cross-correlation coefficient (CCF) data was proposed - in order to investigate and observe the changes of transition postural control strategies organization between COM-COP and ankle-hip angle towards difference translation perturbation order of frequency given.
• Broad comments
Introducing overview was supported by methodology used to fulfil aim of the experiment. Experimental design was appropriate, followed and supported by relevant and precise conclusions. However, explanation of limitations of the study were stated (only observed on the ankle and hip strategies since these strategies were mainly used in postural control), but without closure of space for further discussion/debate (e.g., spine scoliosis, even minor, often influences ankle-hip angle, therefore Adams test, or Cobb’s angles, etc. might have been used to fortify selection criteria within sample of participants).
• Specific comments
1. Minor spell check required
2. Line 137-140: There are sufficient number of official kinematic protocols (Davis protocol, or else) standardized within IBP (International Biological Program), so next text is correct - “motion of the passive marker attached over the joints of the subjects. A fix 16 reflective markers (placed at 3rd metatarsal, lateral malleolus, lateral condyle, trochanter of the femur, iliac crest, acromion of spacula, top of the head and four markers on the force plate) were attached over the subjects’ joints. Both right and left knee were locked to prevent bias movement at the knees during all experimental trials.” – however, please name the standard under which motion was registered

Author Response
Broad comments: Introducing overview was supported by methodology used to fulfil aim of the experiment. Experimental design was appropriate, followed and supported by relevant and precise conclusions. However, explanation of limitations of the study were stated (only observed on the ankle and hip strategies since these strategies were mainly used in postural control), but without closure of space for further discussion/debate (e.g., spine scoliosis, even minor, often influences ankle-hip angle, therefore Adams test, or Cobb’s angles, etc. might have been used to fortify selection criteria within sample of participants).
Response 1:
· Further research will be needed to investigate the postural strategy coordination transition with the presence of knee joint where we believed the multi-segmented body movement could give as deeper understanding in postural control strategy coordination. (Line 365-368)
Specific comments:
Point 1: Minor spell check required
Response 1: Minor spell check have been fulfilled.
Point 2: Line 137-140: There are sufficient number of official kinematic protocols (Davis protocol, or else) standardized within IBP (International Biological Program), so next text is correct - “motion of the passive marker attached over the joints of the subjects. A fix 16 reflective markers (placed at 3rd metatarsal, lateral malleolus, lateral condyle, trochanter of the femur, iliac crest, acromion of spacula, top of the head and four markers on the force plate) were attached over the subjects’ joints. Both right and left knee were locked to prevent bias movement at the knees during all experimental trials.” – however, please name the standard under which motion was registered
Response 2: In this paper, we modified the number of markers set according to Helen Hayes Hospital markers set.

Round 2
Reviewer 1 Report
The authors have addressed each comment that was raised after the initial submission. However, the significance of the research question is extremely unclear, therefore this study have the potential to not contribute to the current literature in postural behaviour. The inconsistency throughout this manuscript, especially in sample size, appropriate data collection and statistical analysis, renders the results and discussion of this study less meaningful.
1. "Based on previous studies, eleven healthy subjects are regarded sufficient, which can provide difference pattern of cross-correlation distribution." Unfortunately they are insufficient. Please provide all parameters for the sample size calculation in the Methods.
2. Unfortunately "Line 118-120" is insufficient. Though studies focused on postural sway, neurological disorders, balance disorder, and vestibular function disorders is another important contributor to postural control. In this study, no exclusion or inclusion criteria about these function or disorders was stated.
3. "The data in this paper is not scattered data/random data where the data were based on frequency/time domain that is steadily increased and decreased. Therefore, the normal distribution of data is not needed to be confirmed." The clarification is insufficient. Also, I guess the data were tested with Levene's test for homogeneity of variance, if the MATLAB software was used? Perhaps using the nonparametric Mann-Whitney U Test instead. Please clarify.
4. "This analysis gave no effect in the transition phase of postural strategy. Therefore, we decide to remove this data from the paper." This is not sufficient. The statistical analysis is vaguely described, and the remove of two-way ANOVA analysis is inadequacy.
5. Line 322-329: These Discussion might not be accurate in light of the results.
6. CONCLUSIONS
Although clearly stated, these conclusions are not accurately representative of the results of this study.
7. Consider adding limitations after 'Discussion'.
8. Line 366-367: remove “The sigmoid function model based on CCF coefficient technique may be a new reliable method in investigating the transition of postural strategy coordination.”. As this study was not longitudinal.
Author Response
Please see the attachment below

Round 3
Reviewer 1 Report
The authors have addressed each comment that was raised after the second submission. However, this manuscript has serious methodological deficiencies.
1. "Based on the reviewer suggestion, sample size was calculated by using below equation. Confidence interval of 95% were used in this calculation which provide z score of 1.96." Sample size reporting is imperfect. The calculation of sample size is not interval of 95% and z score of 1.96. Were power calculations performed a priori?
2. "In this study, transition point was determined by using sigmoid model based on crosscorrelation function (CCF) coefficient data. Based on the reviewer suggestion, CCF coefficient data was performed with Levene’s test for homogeneity of variance." See previous comments.Statistical reporting is imperfect. Were the data statistically tested for normal distribution?
3. What is the evidence that concluded that the transition frequency of postural control strategies for both the COM-COP and ankle-hip data of apparent between 0.4 Hz and 0.6 Hz?